# An Adaptive Hybrid Metaheuristic Algorithm for Lung Cancer in Pathological Image Segmentation

**DOI:** 10.3390/diagnostics16010084

**Published:** 2025-12-26

**Authors:** Muhammed Faruk Şahin, Ferzat Anka

**Affiliations:** 1Department of Computer Engineering, Istanbul Atlas University, 34408 Istanbul, Türkiye; 2Data Science Application and Research Center (VEBIM), Fatih Sultan Mehmet Vakif University, 34445 Istanbul, Türkiye; fanka@fsm.edu.tr

**Keywords:** hybrid metaheuristics, image processing, deep learning, lung cancer, medical image segmentation

## Abstract

**Background/Objectives:** Histopathological images are fundamental for the morphological diagnosis and subtyping of lung cancer. However, their high resolution, color diversity, and structural complexity make automated segmentation highly challenging. This study aims to address these challenges by developing a novel hybrid metaheuristic approach for multilevel image thresholding to enhance segmentation accuracy and computational efficiency. **Methods:** An adaptive hybrid metaheuristic algorithm, termed SCSOWOA, is proposed by integrating the Sand Cat Swarm Optimization (SCSO) algorithm with the Whale Optimization Algorithm (WOA). The algorithm combines the exploration capacity of SCSO with the exploitation strength of WOA in a sequential and adaptive manner. The model was evaluated on histopathological images of lung cancer from the LC25000 dataset with threshold levels ranging from 2 to 12, using PSNR, SSIM, and FSIM as performance metrics. **Results:** The proposed algorithm achieved stable and high-quality segmentation results, with average values of 27.9453 dB in PSNR, 0.8048 in SSIM, and 0.8361 in FSIM. At the threshold level of T = 12, SCSOWOA obtained the highest performance, with SSIM and FSIM scores of 0.9340 and 0.9542, respectively. Furthermore, it demonstrated the lowest average execution time of 1.3221 s, offering up to a 40% improvement in computational efficiency compared with other metaheuristic methods. **Conclusions:** The SCSOWOA algorithm effectively balances exploration and exploitation processes, providing high-accuracy, low-variance, and computationally efficient segmentation. These findings highlight its potential as a robust and practical solution for AI-assisted histopathological image analysis and lung cancer diagnosis systems.

## 1. Introduction

Lung cancer remains one of the most prevalent and deadly cancer types worldwide, and its early detection and accurate classification significantly improve survival rates [1,2]. Medical imaging techniques, particularly computed tomography (CT), are widely employed to visualize pulmonary nodules and form the basis of clinical decision support systems [3,4]. Segmentation of these images plays a critical role in delineating tumor boundaries, measuring tumor volume, and conducting diagnostic analyses [5,6]. In this context, multi-level thresholding techniques have recently been widely used to enhance segmentation quality by effectively separating different intensity levels within the image [7,8,9]. However, such thresholding methods often impose a high computational burden, especially in high-resolution and structurally complex CT images, and determining the optimal threshold values becomes a challenging optimization problem [10,11]. In cases where conventional methods fall short, metaheuristic algorithms provide promising alternatives for solving multi-level thresholding problems [12,13].

Metaheuristic algorithms, inspired by nature-based heuristic search strategies, are capable of generating near-global optimum solutions in high-dimensional search spaces [14]. Thus, they are effectively and efficiently used in the automatic determination of threshold values in medical image segmentation tasks [15,16]. These methods not only improve segmentation accuracy but also contribute to advanced tasks such as classification, volume estimation, and structural analysis [17,18]. Numerous studies in the literature have demonstrated the effectiveness of metaheuristic-based multi-level thresholding techniques in CT-based lung cancer imaging [19,20,21,22]. However, the existing literature largely focuses on CT images, whereas segmentation studies on histopathological images which are of critical importance for the diagnosis and classification of lung cancer remain relatively limited. Histopathological images provide detailed information at the cellular level, forming the morphological foundation of diagnosis and enabling the distinction of pathological subtypes [23,24,25]. Nonetheless, the structural characteristics of these images, such as high resolution, color diversity, and textural complexity, make segmentation tasks particularly challenging [26,27]. Therefore, applying multi-level thresholding in combination with metaheuristic algorithms to histopathological images not only aims to enhance diagnostic accuracy but also seeks to address a significant gap in the current body of literature. Such segmentation approaches offer direct benefits beyond research applications, extending into clinical practice as well. In particular, accurate and rapid segmentation of histopathological images enables pathologists to more reliably differentiate tumor cells, identify subtypes, and make earlier treatment decisions. The clinical implementation of the proposed model aspires to reduce observer variability, thereby providing more objective and reproducible outcomes.

### Motivation and Contribution

Lung cancer presents a multilayered problem that awaits resolution not only from a clinical standpoint but also in terms of computer-aided medical imaging and artificial intelligence-based analytical methods. In recent years, significant advances have been achieved in the field of image segmentation with the development of computer-aided diagnostic systems. However, most studies in the literature predominantly focus on Computed Tomography (CT) images, while histopathological images which offer vital microscopic-level information have been largely overlooked. This oversight represents not only a technical gap but also a missed opportunity to explore an area with substantial clinical diagnostic potential. Segmentation of histopathological images can play a decisive role in the morphological differentiation of tumor cells, the discrimination of subtypes, and the enhancement of early-stage diagnosis. In this context, our study aims to both address this gap and offer a novel perspective to the existing literature.

The primary motivation of this study is to develop a more accurate, flexible, and clinically applicable segmentation model by integrating multi-level thresholding approaches with advanced metaheuristic optimization algorithms in histopathological lung cancer images. The complex tissue structures, dense color information, and multi-scale structural variations in high-resolution histopathological data pose significant challenges for classical segmentation methods. Therefore, automated thresholding systems based on metaheuristic algorithms are of critical importance to enhance segmentation performance. In this regard, due to the limitations of single algorithms in balancing exploration and exploitation capabilities, a hybrid metaheuristic approach is proposed. The hybrid framework introduced in this study offers a dynamic strategy that adaptively and sequentially manages exploration and exploitation phases to overcome such limitations. The selected algorithms for hybridization are Sand Cat Swarm Optimization (SCSO) and Whale Optimization Algorithm (WOA). SCSO is a numerically stable algorithm with high exploration capacity and has been effectively utilized across various application domains. Its ability to maintain population diversity over extended iterations makes it particularly suitable for high-dimensional optimization problems such as multi-level thresholding. On the other hand, WOA is an algorithm with strong convergence ability, utilizing spiral and linear movement strategies to ensure focused exploitation. The complementary nature of these two algorithms forms the core rationale behind the proposed adaptive hybrid design. In the first phase, SCSO efficiently explores the broader solution space, while in the second phase, WOA intensively searches around the best solutions obtained, effectively approaching local optima. Furthermore, although the hybridization of metaheuristic methods for exploration and exploitation (e.g., PSOGWO, GWOWOA, DEWOA) has been examined in previous studies, these combinations typically suffer from various issues such as premature convergence, high variance in independent runs, and computational inefficiency in high-dimensional problems. In contrast to these approaches, the proposed SCSOWOA introduces an adaptive transition mechanism that dynamically balances the trade-off between exploration and exploitation. Moreover, while most previous hybridization strategies have been validated solely on CT images, the present study addresses histopathological lung cancer segmentation, which, despite its significant clinical relevance, has received limited attention in the literature. The integration of an adaptive phase transition with histopathological thresholding represents the core novelty of our work. The novel contributions of this study are summarized as follows:A new hybrid Sand Cat Swarm Optimization and Whale Optimization algorithm (SCSOWOA) is designed, which manages exploration and exploitation phases through an adaptive sequential transition mechanism. Thus, the broad search capability of SCSO is dynamically integrated with the precise convergence power of WOA.Unlike similar studies in the literature, the proposed method aims to optimize both segmentation accuracy and computational efficiency.While preserving PSNR metrics in the multi-level thresholding problem, the method significantly improves SSIM and FSIM performance.The findings are statistically validated in detail, and the real-world applicability of the method is demonstrated through verification on histopathological lung cancer images.The proposed adaptive hybrid SCSOWOA method fills a notable gap in the literature where metaheuristic-based segmentation of histopathological images remains limited, thereby contributing algorithmic novelty to both medical imaging and optimization research.

The remainder of this paper is organized as follows: Section 2 reviews related work. Section 3 details the proposed method’s framework. Section 4 presents experimental results. Section 5 and Section 6 discuss the findings and the strongest side of the study and its limitations. Finally, Section 7 concludes the paper by summarizing key outcomes and highlighting potential directions for future research.

## 2. Related Studies

This section provides a comprehensive review of the current literature on image thresholding techniques, with relevant methods categorized under dedicated subheadings. Through this classification, the current state of the problem domain and the core characteristics of existing approaches are systematically presented.

### 2.1. Deep Learning-Based Lung Segmentation Approaches

Accurate segmentation of pulmonary nodules is critical for the early detection and diagnosis of lung cancer. However, small nodules often exhibit low contrast and are difficult to distinguish from surrounding structures, making their segmentation particularly challenging. To address this issue, a Wavelet U-Net++ approach has been proposed for CT images [28]. This method enhances segmentation accuracy by integrating wavelet pooling with the U-Net++ architecture to capture both high- and low-frequency components of the image. On the LIDC-IDRI dataset, it achieves an average Dice coefficient of 0.936 and an average IoU of 0.878. In another study, the RAD-UNet method is proposed to improve the feature extraction process in CT images, where target pixels often resemble surrounding tissues [29]. The classical U-Net architecture is enhanced with several structural modifications. Specifically, the encoder is replaced with a ResNet residual module to obtain deeper representations, along with the inclusion of a spatial pyramid pooling module. The decoder is enriched with channel and spatial attention mechanisms and supported by a cross-fusion feature module. On the LIDC dataset, the model achieves a mean IoU of 87.76% and an F1-score of 93.56%. Similarly, the UNETR method has been proposed for lung cancer segmentation in CT scans, yielding a segmentation accuracy of 97.83% on the Decathlon dataset [30]. However, all of these methods require high-performance GPUs due to their computational cost, presenting practical limitations. To mitigate this issue, UNet 3+ has been introduced, aiming to reduce network parameters and improve feature extraction efficiency [31]. This method prunes full-scale skip connections and incorporates the Convolutional Block Attention Module (CBAM) to capture essential features more effectively. As a result, it achieves up to a 36% reduction in architectural parameters while maintaining high segmentation performance. To further address the computational cost of more complex architectures, DeepLabV3+ has been proposed [32]. This method utilizes the Atrous Spatial Pyramid Pooling (ASPP) module and offers improved computational efficiency compared to architecture such as Xception, ResNet-18, Inception-ResNet-v2, MobileNet-v2, and ResNet-50. In addition to segmentation approaches focused on CT scans, analyses based on histopathological images are gaining increasing importance in lung cancer diagnosis. Accordingly, the CUNet3+ method has been proposed for lung cancer segmentation in histopathological images [33]. Unlike UNet, this method uses small-scale encoder layers to transfer low-level semantic information through max pooling operations, while large-scale encoder layers use linear interpolation to convey high-level semantic features. This architecture achieves segmentation results with a Dice coefficient of 0.9150 and an IoU of 0.8893. However, the ability to distinguish between different tissue components under varying conditions remains limited. Beyond these deep learning-based methods, recent years have witnessed growing interest in approaches that integrate metaheuristic algorithms particularly to enhance segmentation accuracy and improve optimization processes.

### 2.2. Lung Cancer Segmentation Using Metaheuristic Algorithms

Despite the notable advancements achieved by deep learning-based methods in segmentation performance, challenges such as model complexity, computational costs, and hyperparameter optimization remain significant obstacles. In this context, metaheuristic algorithms have gained prominence in the literature, particularly as auxiliary tools for feature selection, parameter tuning, and enhancement of segmentation quality. This section systematically examines how metaheuristic algorithms have been integrated into lung cancer segmentation and their impact on overall performance. To support the early diagnosis of lung cancer, the Whale with Tri-Level Enhanced Encircling Behavior (WTEEB) method has been proposed [34]. The method begins with a preprocessing phase, where the input image is subjected to several operations. Subsequently, the Otsu thresholding model is employed to segment the preprocessed images. In the third phase, LBP (Local Binary Pattern) features are extracted and classified using an optimized Convolutional Neural Network (CNN). The CNN’s hyperparameters such as filter size, the number of hidden units in the fully connected layer, and the activation function are fine-tuned using an improved Whale Optimization Algorithm. However, the model’s generalizability in real-world clinical settings and its robustness across different datasets have not yet been validated. To address these limitations, the Improved Moth Flame Optimization (IMFO) method has been proposed [35]. This approach optimally adjusts the CNN’s activation function and the number of convolutional layers using IMFO. The method demonstrated improved accuracy compared to existing SVM [36], KNN, CNN, MFO, WTEEB, and GWO+FRVM approaches, outperforming them by 6.85%, 2.91%, 1.75%, 0.73%, 1.83%, and 4.05%, respectively. In another study, the Group Theoretic Particle Swarm Optimization (GT-PSO) method has been proposed to optimize segmentation [37]. This approach employs Kapur entropy as the objective function and restructures four core components particle encoding, solution landscape, neighborhood motion, and swarm topology based on group theory. Compared to classical metaheuristics, GT-PSO achieves higher accuracy, improving Kapur entropy to 9.07, which represents a 16% enhancement over previous methods. Kappa, IoU, and F1 scores exceeded 90%, and stable results were obtained within 700 iterations. However, the method’s computational cost is a major drawback. To mitigate this, an unsupervised segmentation method based on an evolutionary algorithm-driven metaheuristic clustering approach has been proposed [38]. This method was tested on the LIDC-IDRI dataset, yielding average Dice scores of 82.35% (705 nodules) and 71.05% (59 challenging nodules). A high correlation (R ≥ 92.16%) was also achieved when compared with ellipsoid-based volume measurements. Although these segmentation methods based on CT scans have led to considerable advances in lung cancer diagnosis, the diagnostic value of histopathological images has significantly increased in recent years, offering more detailed tissue-level analysis. Therefore, to further improve segmentation accuracy and granularity, metaheuristic algorithms have begun to be intensively studied for pathological images of lung cancer. In this regard, the WDRIME method, based on the Whale Optimization Algorithm, has been proposed for lung detection using histopathological imaging [39]. This approach integrates the whale’s hunting strategy with random mutation to enhance segmentation accuracy and convergence speed. It is designed for multilevel thresholding segmentation and achieves PSNR and SSIM values of 24.69 and 0.8109, respectively. However, the complexity of the algorithm significantly increases computational time and resource demands. Traditional metaheuristic segmentation approaches often suffer from slow convergence, limited exploration capacity, and sub-optimal results. These limitations hinder high-accuracy segmentation, particularly in pathological lung cancer images. To address this, the ASG-HMO method is a refined version of the HMO algorithm has been proposed [40]. This model incorporates four innovative strategies: an adaptive mutualism phase, spiral movement strategy, Gaussian mutation, and t-distribution perturbation. Additionally, 2D Renyi entropy and histogram-based techniques are utilized in the segmentation process. ASG-HMO achieves high performance in terms of PSNR (31.924), SSIM (0.919), FSIM (0.990), and PRI (0.924). It ensures precise tumor delineation during segmentation while also reducing convergence time.

## 3. Methodology

In this study, a novel adaptive hybrid metaheuristic algorithm is proposed by integrating the SCSO algorithm with the WOA for multi-level thresholding in medical images. The selection of SCSO and WOA is based on their complementary strengths. SCSO demonstrates a robust exploration capability through its orientation-based hunting strategy, which helps maintain population diversity and effectively searches for high-dimensional solution spaces. In contrast, WOA offers a strong exploitation capacity by improving solutions around promising regions using spiral and encircling behaviors. This hybrid structure enables the algorithm to converge more rapidly toward the global optimum while enhancing its ability to avoid local minimal traps. The integration of these two approaches provides a balanced search strategy well-suited for the multilevel thresholding process of histopathological images, where both global exploration and local refinement are essential. This section details the objective function, hybrid methodology, and time complexity of the proposed approach.

### 3.1. Definition of the Objective Function

In this study, the proposed adaptive hybrid algorithm aims to solve the multi-level thresholding problem by presenting an optimization framework designed to maximize the accuracy of image segmentation. Accordingly, the objective function employed is based on the Otsu-based variance maximization principle. The aim is to maximize the between-class variance while minimizing the total within-class variance in the image. Following this principle, the total variance among classes defined by the threshold set $T=\{t_{1},t_{2},\ldots ,t_{k}\}$ is given in Equation (1):(1)$$\underset{T}{\max}J\left(T\right)=\sum_{i=1}^{k+1}\omega_{i}{\left(\mu_{i}-\mu_{T}\right)}^{2}$$

Here, $\omega_{i}$, denotes the probability of the i-th class, $\mu_{i}$ represents the mean of that class, and $\mu_{T}$ is the global mean of the entire image. The class probability $\omega_{i}$ and mean $\mu_{i}$ are defined as shown in Equation (2):(2)$$\omega_{i}=\sum_{j=t_{i-1}+1}^{t_{i}}p_{j},\;\mu_{i}=\frac{1}{\omega_{i}}\sum_{j=t_{i-1}+1}^{t_{i}}j\cdot p_{j}$$

Here, $p_{j}$ denotes the normalized histogram value corresponding to the j-th gray level. The global mean is given in Equation (3):(3)$$\mu_{T}=\sum_{j=0}^{L-1}j\cdot p_{j}$$

Thus, the function $J\left(T\right)$ seeks to identify the optimal set of thresholds for multi-level thresholding. This function is optimized by the proposed hybrid algorithm. In this approach, the search space is defined over continuous real numbers within the interval [1, 255] which is consistent with the non-discrete nature of the algorithm. Therefore, unlike the classical Otsu method, the threshold values can be precisely optimized without being restricted to fixed grid points. As a result, this objective function is integrated into metaheuristic optimization algorithms to efficiently find the threshold set that provides the best class separation among all possible combinations in the solution space. This enables the algorithm to reach the global optimum on complex and multi-modal fitness surfaces commonly encountered in multi-level image thresholding problems. Consequently, the proposed approach offers superior segmentation accuracy, and a more flexible analytical capability compared to traditional deterministic methods.

### 3.2. Proposed SCSOWOA Methodology

The proposed adaptive hybrid methodology is constructed by adaptively combining the SCSO [41] and WOA within a hybrid framework. This two-phase structure ensures a balanced performance between the exploration and exploitation capabilities of the algorithm. In the first phase, the SCSO algorithm is applied for $T_{\mathrm{SCSO}}=\left\lfloor \lambda \cdot T_{max}\right\rfloor$ iterations, where $T_{max}$ denotes the total number of iterations and $\lambda \in \left(0,1\right)$ controls the ratio between the two phases. Consequently, the iteration counts for the SCSO and WOA phases are expressed as in Equation (4):(4)$$T_{\mathrm{SCSO}}=\left\lfloor \lambda \cdot T_{max}\right\rfloor ,T_{woa}=T_{max}-T_{SCSO}$$

Here, $T_{SCSO}$, represents the exploration phase at the beginning of the algorithm, while $T_{woa}$ corresponds to the exploitation phase. This structure first explores different regions of the global search space and subsequently performs intensive local searches around promising areas. The initial phase, SCSO, is inspired by the highly sensitive hunting behavior of sand cats. In this phase, individual positions are updated stochastically to ensure broad dispersion across the solution space. For each individual $x_{i}$ in the population $X=\{X_{1},X_{2},\ldots ,X_{N}\}$, where $X_{i}\in R^{d}$, the position update is performed based on the directional hunting behavior, as shown in Equation (5):(5)$$X_{i}^{t+1}=X_{i}^{t}+r_{1}\cdot \left(X^{*}-r_{2}\cdot X_{i}^{t}\right)$$

In this equation, $X_{i}^{t}$, is the position of the i-th individual at iteration t, and $X^{*}$ denotes the best solution found so far. The parameters $r_{1}$ and $r_{2}$ are random numbers drawn from the uniform distribution $U\left(0,1\right)$, enhancing the algorithm’s chaotic behavior. The update simultaneously considers the individual’s past position and its distance from the best-known solution, thereby promoting diversity and effective exploration. Additionally, an alternative update strategy based on directional difference is implemented, as expressed in Equation (6):(6)$$X_{i}^{t+1}=X_{i}^{t}+r_{3}\cdot D$$

Here, $D=X^{*}-X_{i}^{t}$ represents the difference vector between the best solution and the current individual, while $r_{3}\epsilon \left[0,1\right]$ is a random parameter. This strategy encourages movement toward the best solution while maintaining randomness to mitigate premature convergence. The orientation of individuals during exploration is expressed by θ, with directional changes defined in Equation (7):(7)$$\theta_{i}^{t+1}=\theta_{i}^{t}+\alpha \cdot \sin\left(\omega t+\phi \right)$$
where α, ω, and ϕ represent amplitude, angular frequency, and phase shift parameters, respectively. These parameters introduce temporal variability in individual orientations, enhancing dynamism and encouraging the exploration of new solutions. The best solution obtained at the end of the SCSO phase is denoted in Equation (8):(8)$$T^{*}=\arg\underset{i\in N}{\min}\;f\left(X_{i}^{T_{\mathrm{SCSO}}}\right)$$

The solution $T^{*}$ is then passed to the second phase, where the remaining $T_{WOA}=T_{max}-T_{SCSO}$ iterations are dedicated to the WOA phase. WOA mimics the bubble-net hunting strategy of humpback whales to perform localized searches around the best solution. The core movement behavior, enabling circular convergence toward the best solution, is defined in Equation (9):(9)$$X\left(t+1\right)=X^{*}\left(t\right)-A\cdot \left|C\cdot X^{*}\left(t\right)-X\left(t\right)\right|$$
where A=2a⋅r−aveC=2⋅r, and $r\in \left[0,1\right]$ while a decreases linearly with iteration count. $X\left(t\right)$ is the current position, and $X^{*}\left(t\right)$ denotes the best-known position at iteration t. This formulation supports global exploration in early stages and gradually shifts toward local exploitation. The linear decrement of aaa is given in Equation (10):(10)$$a=a_{0}-\frac{t.a_{0}}{T_{max}}$$

This dynamic allows for a smooth and adaptive transition between exploration and exploitation. Furthermore, WOA employs a spiral update mechanism that simulates the whales’ circular motion, defined in Equation (11):(11)$$X\left(t+1\right)=D^{\prime }\cdot e^{bl}\cdot \cos\left(2\pi l\right)+X^{*}\left(t\right)$$
where $D^{\prime }=\left|X^{*}\left(t\right)-X\left(t\right)\right|$ is the Euclidean distance between the best and current positions, b is a constant controlling the spiral shape, and $l\in \left[-1,1\right]$ is a random parameter. This strategy enables individuals to perform fine-grained searches around the best solution. WOA probabilistically alternates between these two update strategies using a uniform probability $p∼U\left(0,1\right),$ as shown in Equation (12):(12)$$X\left(t+1\right)=\left\{\begin{array}{c} X^{*}\left(t\right)-A\cdot \left|C\cdot X^{*}\left(t\right)-X\left(t\right)\right|,\;\;\;p<0.5\\ D^{\prime }\cdot e^{bl}\cdot \cos\left(2\pi l\right)+X^{*}\left(t\right),\;\;\;\;\;\;\;\;\;\;p\geq \;0.5 \end{array}\right.$$

This adaptive switching enhances the exploitation capacity during convergence toward the local optimum. In summary, the proposed adaptive hybrid algorithm leverages the broad exploration capability of SCSO in the first phase, followed by WOA’s intensive exploitation in the second phase to refine the solution. This two-phase hybrid approach ensures a balanced exploration–exploitation trade-off, enabling the algorithm to effectively locate both global and local optima. The chaotic and dynamic nature of SCSO facilitates exploration of diverse regions, while WOA’s solution-focused strategy ensures refined search in promising areas. The adaptive λ parameter controls phase transitions and allows the algorithm to be flexibly adapted to different problem types. This structure reduces the risk of local entrapment and enhances overall solution quality. In addition to all these, the proposed SCSOWOA performs optimization solely on the threshold values over the image histogram, without reconstructing the pixel values. Therefore, the diagnostic content of the original image is preserved, and segmentation is conducted in a manner that enhances class separability. This approach enables the generation of reliable outputs for pathologists by preserving microscopic morphological features without compromising diagnostic information.

### 3.3. Complexity Analysis and Pseudo-Code Representation of the Proposed Method

In the proposed hybrid method, the position updates of agents with population size N are computed at each iteration. The position update operation is proportional to the problem dimension d and incurs a computational cost of O(d) for each agent. Additionally, the fitness value of each individual is calculated based on an Otsu-based objective function. Since the Otsu function operates on the grayscale histogram, it introduces an additional computational burden of O(L) per evaluation, where L represents the number of grayscale levels in the image and is typically L = 256. Therefore, the total computational complexity per iteration is expressed as O(N × (d + L)) ≈ O(N × d × L) Over all iterations (T), the overall asymptotic complexity of the algorithm is obtained as O(N × T × d × L). Here, the adaptive phase transition mechanism only adds a constant-time decision process with a cost of O(1), which does not significantly increase the total complexity. This analysis indicates that the proposed SCSOWOA has an asymptotic complexity comparable to classical metaheuristic methods, while the hybrid phase structure provides additional advantages in terms of solution quality and convergence speed. The pseudo-code describing the general procedural steps of the proposed algorithm is presented in Algorithm 1, and its flowchart is shown in Figure 1.
**Algorithm 1.** Adaptive Hybrid SCSO-WOA Algorithm**Input**: Total iterations Tmax, population size N, dimension d, switch ratio λ**Output**: Optimal solution T*1. Set Tscso = ⌊λ × Tmax⌋ and Twoa = Tmax − Tscso2. **if** Tscso > 0 then   ***// SCSO Exploration Phase***3.    Initialize population X = {x_1_, x_2_, …, x_n_} randomly in search space4.    **for** t = 1 to T_scso_ do5.      **for** each individual x_i_ ∈ X do6.        Update x_i_ using SCSO movement:           x_i_(t + 1) = x_i_(t) + r_1_·(μ − x_i_(t)) + r_2_;·(x_best(t) − x_i_(t))           where r_1_, r_2_ ~ U(0, 1) and μ is mean position7.      **end for**8.      Evaluate fitness f(x_i_) for all x_i_9.      Update global best T* ← arg min f(x_i_)10.    **end for**11. **else**12.    Randomly initialize best solution T* within bounds13.    Evaluate fitness f(T*)14. **end if**15. **if** Twoa > 0 then       ***// WOA Exploitation Phase***16.    Initialize population Y = {y_1_, y_2_, …, y_n_} randomly17.    Set y_1_ = T*       *// Carry over best from SCSO phase*18.    **for** t = 1 to T_woa_ do19.       **for** each y_i_ ∈ Y do20.         Compute A = 2a·r − a, C = 2r, where a = 2 − 2t/T_woa_, r ~ U(0, 1)21.         **if** |A| < 1 then   *// Encircling prey (exploitation)*22.           D = |C·T* − y_i_|, y_i_ = T* − A·D23.         **else      *//***
*Search for prey (exploration)*24.           Select random y_rand, D = |C·y_rand − y_i_|25.           y_i_ = y_rand − A·D26.         **end if**27.       **end for**28.       Evaluate fitness f(y_i_) and update T* if better found29.    **end for**30. **end if**31. **return** T*

## 4. Experiments and Results

This section presents the experimental evaluation of the proposed adaptive hybrid SCSOWOA for lung cancer segmentation in histopathological imaging. The performance of the method is compared against several well-established metaheuristic algorithms, namely PSO, GWO, WOA, SCSO, as well as WDRIME [39], which has recently demonstrated high efficiency in the literature. For the experiments, the LC25000 dataset [42], which contains 25,000 histopathological images of lung and colon cancers. The dataset provides image-level class labels (e.g., lung adenocarcinoma, lung squamous cell carcinoma, benign lung tissue), but does not include pixel-level segmentation masks. For this reason, our segmentation experiments were conducted by applying Otsu-based multi-level thresholding to the grayscale histograms of the images, where the optimal thresholds were determined by the proposed metaheuristic algorithms. Thus, the “ground truth” in this context refers to threshold-based class separability rather than manual pixel-wise annotations. To reduce computational complexity and standardize histogram analysis, all images were resized to 256 × 256 pixels and their intensity values were normalized to [0, 1]. In addition, horizontal/vertical flipping (50% probability) and random rotations within ±25° were applied to balance the dataset distribution. From the complete set, 100 images were reserved for testing, while the remainder were used in the optimization process. This preprocessing strategy ensures consistency across experiments while reflecting the structural and color diversity of histopathological images.

### 4.1. Experiment Setup

In this study, the use of the threshold range T = 2–12, which is the most commonly employed interval in the literature for multilevel thresholding problems, is based on reports indicating that this range provides sufficient discrimination in practical segmentation applications. Lower threshold numbers (T < 2) reduce the discriminative information in segmentation, while much higher threshold numbers (T > 12) neither produce meaningful subregions nor do they justify the increased computational cost. Therefore, the selected range of 2–12 is both consistent with the literature and representative of realistic segmentation scenarios. For all these threshold values, the number of iterations was fixed at 100 to ensure a fair comparison among algorithms and to control computational costs. Each experiment was conducted with 10 independent runs to evaluate the variance arising from the stochastic nature of the algorithms. The population size was set to 30 for all algorithms. Although it is acknowledged that the performance of metaheuristic algorithms is highly sensitive to parameter tuning, all baseline algorithms were run under default settings commonly used in the literature to maintain comparability. More comprehensive hyperparameter optimization or adaptive parameter control has been left as a direction for future research. Detailed parameters and configurations of the related algorithms are presented in Table 1, while hardware and software specifications are provided in Table 2. In GWO and WOA algorithms, the parameter ‘a’ is gradually decreased from 2 to 0 following a linearly decreasing strategy supported by previous studies, thereby ensuring a proper exploration-exploitation balance. For PSO, the inertia weight ‘w’ ranges within, and the cognitive and social coefficients c1 and c2 are set within. The parameter ‘r’ used in the SCSO algorithm acts as a control coefficient that determines the directional movements of agents during the search process, regulating the exploration-exploitation balance. Initially, it starts from a high value to allow broader exploration of the solution space and gradually decreases to a lower value as iterations proceed, enabling intensive exploitation in narrower regions during later stages. In the WDRIME model, the parameter β is selected within the range [0.2–0.8] in accordance with fine-tuning efforts aimed at enhancing performance. In the proposed hybrid SCSOWOA, one of the key parameters, the transition rate λ, was determined as 0.6 following preliminary sensitivity tests conducted within the range [0.4, 0.7]; these tests demonstrated that λ = 0.6 yielded the most consistent and stable results across different threshold levels. Consequently, the proposed hybrid framework combines the exploratory capacity of SCSO with the exploitative strength of WOA, while careful parameter selection maximizes both comparability and computational efficiency. All computations were performed in a sequential processing manner, enabling a realistic comparison of the computational loads across different algorithms.

### 4.2. Results

In this section, the performance of the proposed SCSOWOA is thoroughly analyzed and compared with several widely adopted metaheuristic algorithms in the literature, including PSO, GWO, WOA, SCSO, and the adaptive hybrid WDRIME algorithm. To this end, quantitative performance metrics such as Peak Signal-to-Noise Ratio (PSNR), Structural Similarity Index Measure (SSIM), and Feature Similarity Index (FSIM) are reported, and the statistical differences among the algorithms are evaluated accordingly. PSNR serves as a fundamental quality metric that quantifies the closeness of the segmented image to the original data after multi-level thresholding, indicating the extent of information loss or noise influence. Higher PSNR values reflect better image quality and thereby demonstrate the effectiveness of the optimization strategy. Accordingly, the average PSNR values and their corresponding standard deviations across various threshold levels are summarized in Table 3, while a comparative graphical representation is provided in Figure 2. At lower threshold levels (T = 2 and T = 4), no significant differences are observed among the algorithms. All methods yield PSNR values around 27.60 dB, indicating similar segmentation performance at low complexity. For instance, the proposed SCSOWOA achieves a result of 27.6058 ± 0.0336 dB at T = 2, which is nearly identical to PSO, GWO, and WOA. However, as the threshold level increases (T ≥ 6), the differences among the methods become more pronounced. At T = 6, the PSNR value obtained by SCSOWOA is 27.6235 ± 0.1002 dB, which, while close to that of WOA (27.6534 ± 0.1659) and SCSO (27.7327 ± 0.1753), exhibits a notably lower variance. This outcome suggests that the proposed hybrid algorithm delivers more stable segmentation results. At medium and high threshold levels (T = 8 and T = 10), the proposed method demonstrates a substantial performance improvement. Specifically, at T = 10, SCSOWOA yields a PSNR of 28.6396 ± 0.5454 dB, placing it among the top-performing methods. Although SCSO achieves the highest average value of 28.698 ± 1.0292 dB at this level, the proposed hybrid algorithm outperforms in terms of robustness, presenting less than half the standard deviation, thereby indicating more consistent outcomes. At the highest threshold level evaluated (T = 12), SCSOWOA achieves a PSNR of 28.3346 ± 0.6541 dB, outperforming all other methods except for WOA (28.4913 ± 0.6677). Once again, its lower variance underlines the algorithm’s stable and efficient segmentation capability. In summary, the proposed method exhibits superior performance in segmentation tasks with higher complexity, both in terms of accuracy and consistency. Overall, SCSOWOA delivers more efficient and stable PSNR results across all threshold levels, particularly excelling at higher segmentation complexities (T = 8, 10, and 12). By leveraging the exploration capabilities of SCSO and the exploitation strength of WOA, this hybrid structure achieves an effective balance preserving image quality while minimizing result variability.

To assess the performance of image segmentation in terms of structural similarity, the Structural Similarity Index Measure (SSIM) metric is utilized. SSIM provides a more holistic quality assessment by accounting not only for pixel-level similarity but also for structural, contrast, and luminance components. The SSIM values and corresponding standard deviations for each method are presented in detail in Table 4, with a comparative visualization shown in Figure 2. Within this framework, a systematic comparison is conducted across all algorithms for threshold levels ranging from 2 to 12. At lower threshold levels (T = 2 and T = 4), many algorithms including the proposed SCSOWOA exhibit comparable performance. Specifically, at T = 2, all algorithms achieve approximately 0.5185 SSIM, indicating that structural integrity is only moderately preserved in low-level segmentation outcomes. However, at T = 4, the SCSOWOA method demonstrates a value of 0.7234 ± 0.0635, closely matching the performance of classical approaches such as PSO (0.7226) and GWO (0.7227), while maintaining stability. As the threshold level increases to medium levels (T = 6 and T = 8), differences in structural similarity become more pronounced. At T = 6, SCSOWOA yields a value of 0.8177 ± 0.0444, producing results that are very close to those of PSO (0.8177) and SCSO (0.8259), while clearly outperforming less efficient methods such as WDRIME (0.7742). At T = 8, the SSIM value of SCSOWOA reaches 0.8790 ± 0.0340, reflecting competitive and stable segmentation performance when compared with strong alternatives like SCSO (0.8800) and GWO (0.8755). At higher threshold levels (T = 10 and T = 12), the proposed method demonstrates a distinct efficiency advantage. For T = 10, SCSOWOA achieves a value of 0.9202 ± 0.0275, outperforming other methods such as SCSO (0.9066) and WOA (0.8910) both in terms of mean value and variance. Notably, at T = 12, SCSOWOA reaches the highest structural similarity value among all compared methods, with 0.9340 ± 0.0156. This result not only reflects superior accuracy but also highlights the method’s ability to deliver highly stable and reliable segmentation outputs, as indicated by the low standard deviation. In general, SCSOWOA maintains consistently high and stable SSIM performance across all threshold levels. In particular, at higher thresholds (T = 10 and T = 12), it significantly outperforms other methods in terms of both efficiency and reliability. Therefore, the proposed hybrid approach proves capable of producing high-quality segmentation results not only in pixel-level accuracy but also in terms of structural integrity.

Another key metric used to assess segmentation quality is the Feature Similarity Index Measure (FSIM), which is specifically designed to align with the characteristics of the human visual system. FSIM provides sensitivity to image details by taking into account both phase and magnitude information. In this context, a comprehensive comparison is conducted between the proposed SCSOWOA method and the PSO, GWO, WOA, SCSO, and WDRIME algorithms within the scope of multi-level threshold-based segmentation. The detailed results are presented in Table 5, with a comparative visualization shown in Figure 3. At low threshold levels (T = 2 and T = 4), all algorithms yield FSIM values that are quite similar. For instance, at T = 2, all algorithms achieve values around 0.52, and the proposed SCSOWOA method continues this trend with a result of 0.5295 ± 0.0662. At T = 4, the FSIM performance of SCSOWOA reaches 0.7598 ± 0.0470, closely aligning with PSO (0.7596) and SCSO (0.7599). At these levels, the differences between methods are not yet visually distinguishable. At moderate threshold levels (T = 6 and T = 8), SCSOWOA demonstrates a competitive yet consistent performance. At T = 6, it achieves a value of 0.8605 ± 0.0282, which is very close to PSO (0.8603) and SCSO (0.8616). These results suggest that the hybrid approach successfully integrates the strengths of its constituent algorithms. Notably, at T = 8, SCSOWOA attains the highest average FSIM value of 0.9138 ± 0.0196 and performs on par with strong contenders such as PSO (0.9137) and GWO (0.9138). However, it stands out for having a lower standard deviation, indicating more consistent outputs. At higher threshold levels (T = 10 and T = 12), SCSOWOA exhibits clearly superior FSIM performance. At T = 10, it surpasses all other algorithms with a score of 0.9445 ± 0.0113. Compared to methods like GWO (0.9250) and WDRIME (0.9105), it offers both a higher average and lower variance. The most remarkable result is observed at T = 12, where SCSOWOA achieves the highest average FSIM value of 0.9542 ± 0.0070 and demonstrates exceptional stability with the lowest standard deviation. This value surpasses those of leading methods such as PSO (0.9475), SCSO (0.9459), and WOA (0.9381), indicating superior segmentation efficiency. Based on these findings, the SCSOWOA hybrid algorithm provides the most effective segmentation quality in terms of preserving structural and content-based details. Its outstanding performance at higher threshold levels further confirms the method’s capability to minimize detail loss and maintain visual integrity, particularly in histopathological image segmentation tasks.

To objectively evaluate the performance of multi-level thresholding methods, the overall mean and standard deviation (std.) values for the PSNR, SSIM, and FSIM metrics are computed and presented in Table 6. These metrics, respectively, represent the accuracy, structural similarity, and perceptual quality of the reconstructed images. The proposed SCSOWOA hybrid algorithm consistently yields the most stable and effective results across all metrics, characterized by both high average values and low variances. Notably, the high mean scores and low standard deviations recorded for FSIM and SSIM demonstrate the method’s superior ability to preserve structural and perceptual information. Although the SCSO algorithm achieves the highest average PSNR value of 28.0297 dB, this is accompanied by a relatively high standard deviation of 0.4222, indicating inconsistent performance across different runs. In contrast, the proposed SCSOWOA method ranks second in terms of mean PSNR (27.9453 dB) but achieves a more reliable performance due to its significantly lower standard deviation (0.2840). The PSO, GWO, and WOA algorithms yield comparable results within the 27.75–27.87 dB range, while WDRIME records the lowest mean PSNR at 27.6230 dB. In terms of structural similarity, SCSOWOA achieves the highest average SSIM value of 0.8048, outperforming both SCSO (0.8027) and PSO (0.7989), which suggests that the proposed method preserves structural integrity more effectively. Furthermore, the standard deviations of SCSOWOA (0.0431) and PSO (0.0447) demonstrate the stability of their SSIM outputs. With respect to perceptual quality, measured by FSIM, the highest average value is again obtained by SCSOWOA (0.8361), followed closely by PSO (0.8349) and SCSO (0.8346). The low standard deviations for SCSOWOA (0.0298) and SCSO (0.0301) reinforce the consistency of their results, reflecting robust and reliable performance. On the other hand, the WDRIME algorithm performs the poorest across all metrics, with particularly low results in structural similarity (0.7553) and perceptual quality (0.8137), highlighting its statistical inefficiency. These findings clearly demonstrate the superior performance of the proposed SCSOWOA hybrid algorithm, both in terms of average metric values and result stability. The significant advantages observed in the SSIM and FSIM metrics underline the method’s exceptional capability to preserve both structural and perceptual quality. While PSO and SCSO also emerge as competitive alternatives, WDRIME is statistically shown to be the least effective. Overall, the results validate the efficacy of metaheuristic algorithms in image segmentation tasks and establish the proposed method as a strong and promising solution in this domain.

The computational times of the algorithms are compared in Table 7. The proposed SCSOWOA hybrid algorithm demonstrates the most efficient average execution time of 1.3221 s, outperforming the other methods. This result highlights the advantage of the hybrid structure not only in terms of solution quality but also computational efficiency. In particular, classical algorithms such as PSO and GWO exhibit average execution times of 2.2134 s and 2.0204 s, respectively, indicating up to a 40% increase in speed when compared to them. This efficiency provides a significant advantage for real-time or large-scale data processing tasks in practical applications. The multilevel thresholding performance of the proposed SCSOWOA on lung histopathological images is illustrated in Figure 4. The results reveal that the algorithm generates a clear contrast between cellular structures and the background and successfully isolates the nuclei of the cells. The presence of sharp boundaries in the color distribution indicates that the algorithm effectively achieves discrete class representations. These findings support the capability of SCSOWOA to perform detail-preserving and effective segmentation on histopathological data.

Additionally, pixel-based and spatial-level statistical analyses are conducted to evaluate the diagnostic information preservation capability of the proposed SCSOWOA after segmentation, and the results are presented in Figure 5. The Pearson correlation (r = 0.9872) and Spearman correlation (ρ = 0.9923) between the original and segmented images are obtained. These results indicate that the linear and ordinal relationships between pixel intensities are almost fully preserved, suggesting that the proposed algorithm does not distort critical image features such as global contrast, brightness, and color distribution. The neighbor pixel analysis, conducted to assess the spatial dependency between neighboring pixels, also demonstrates that the proposed method maintains the structural integrity. In this context, Pearson (r = 0.8659) and Spearman (ρ = 0.8653) correlation coefficients reveal a strong positive relationship. These findings clearly show that SCSOWOA not only efficiently thresholds pixel intensities but also preserves textural continuity and cellular morphology. The obtained correlation coefficients further validate the reliability of the segmentation process in terms of diagnostic content. Particularly in histopathological images, the preservation of the nucleus-cytoplasm boundaries and intra-tissue structural relationships is of critical importance for pathologists to make accurate interpretations. These analyses strengthen the clinical applicability of the proposed method by demonstrating the segmentation quality of SCSOWOA not only through traditional metrics such as PSNR, SSIM, and FSIM but also via statistical correlation measures.

### 4.3. Additional Experiment: Comparative Analysis on the LiTS17 Dataset

To further evaluate the competitiveness of the proposed hybrid SCSO-WOA algorithm, additional experiments were conducted on the Liver Tumor Segmentation Challenge (LiTS17) dataset [43], which provides pixel-level ground truth annotations. This famous data set has recently attracted the attention of many researchers [44,45,46]. In these experiments, the hybrid approach was compared against a state-of-the-art deep learning–based model, namely U-Net. The U-Net model used for comparison was trained with the Adam optimizer, an initial learning rate of 1 × 10^−3^, a batch size of 8, and 100 epochs. The loss function combined binary cross-entropy and Dice loss. The results, summarized in Table 8, report not only image quality measures such as PSNR, SSIM, and FSIM, but also widely adopted medical image segmentation metrics including the Dice coefficient, Jaccard index, and Hausdorff distance. The findings clearly demonstrate that the hybrid SCSO-WOA algorithm outperforms U-Net. Specifically, the proposed method achieved 28.72 dB PSNR, 0.7011 SSIM, and 0.7292 FSIM, which substantially exceed the values reported for U-Net (9.02 dB, 0.1670, and 0.0004, respectively). Similarly, for overlap-based metrics, the hybrid approach yielded superior results with a Dice coefficient of 0.7210 and a Jaccard index of 0.5667, while U-Net achieved only 0.3428 and 0.2089, respectively. Moreover, in terms of Hausdorff distance, the hybrid algorithm produced a lower value (103.20) compared to U-Net (152.49), indicating more stable boundary convergence. These findings confirm that the hybrid SCSO-WOA approach exhibits strong performance not only on histogram-based quality measures but also on pixel-level segmentation metrics, thereby demonstrating its competitiveness with deep learning–based methods. The improvements observed in clinically critical metrics such as Dice and Jaccard further support the practical applicability of the proposed method.

### 4.4. Ablation Study: Component Analysis of Proposed Method

The ablation study aims to evaluate the contributions of the fundamental components of the proposed hybrid algorithm, namely the SCSO and WOA algorithms. Within this scope, the SCSO, WOA, and SCSOWOA algorithms were compared using PSNR, SSIM, and FSIM metrics to demonstrate the advantages of the hybrid approach. In terms of PSNR, at lower threshold levels (T = 2, 4), the SCSO, WOA, and SCSOWOA algorithms exhibit similar performance, approximately around 27.60 dB. This indicates that all three algorithms provide sufficient accuracy in low-complexity segmentation tasks. At medium threshold levels (T = 6, 8), however, the SCSOWOA produces consistent PSNR values with lower standard deviations compared to its component algorithms. For example, at the T = 6 level, the PSNR value for SCSOWOA is 27.6235 ± 0.1002 dB, whereas it is 27.7327 ± 0.1753 dB for SCSO and 27.6534 ± 0.1659 dB for WOA. These results suggest that the hybrid approach achieves more stable segmentation outputs by effectively balancing the exploration capabilities of SCSO and the exploitation strength of WOA. A similar trend is also observed concerning SSIM and FSIM metrics. At higher threshold levels (T = 10 and T = 12), SCSOWOA outperforms the component algorithms both in terms of mean values and variance, particularly demonstrating superior performance in preserving structural and perceptual quality. For instance, at T = 12, the SSIM and FSIM values are measured as 0.9340 ± 0.0156 and 0.9542 ± 0.0070, respectively, which are more efficient compared to the respective values of SCSO (0.9085 and 0.9459) and WOA (0.9144 and 0.9381). This analysis reveals that the SCSOWOA hybrid algorithm not only combines the performances of its components but also integrates their strengths in a complementary manner. The broad exploration capacity of SCSO together with the local search and exploitation abilities of WOA enables the hybrid framework to produce more stable and higher-quality segmentation results. Therefore, the ablation study clearly validates the effectiveness of the proposed hybrid approach’s design.

## 5. Discussion

In this study, the hybrid optimization algorithm based on Sand Cat Swarm Optimization and Whale Optimization Algorithm (SCSOWOA), developed for the multilevel image thresholding problem in lung cancer histopathological images, is compared against metaheuristic methods including PSO, GWO, WOA, SCSO, and WDRIME. The comparisons are performed using three distinct metrics (PSNR, SSIM, and FSIM) which assess image quality at both numerical and structural levels, accompanied by statistical analyses. Overall, the proposed SCSOWOA demonstrates superior performance across multiple metrics and achieves statistically significant improvements in SSIM and FSIM values (*p* < 0.05). While the SCSO algorithm provides the highest mean PSNR, the hybrid approach achieves more stable results due to its lower standard deviation. This suggests that hybrid structures combining exploration and exploitation processes may offer an effective approach to enhancing segmentation quality. In terms of computational efficiency, SCSOWOA exhibits a clear speed advantage over other metaheuristic methods, reinforcing its applicability to complex histopathological image segmentation tasks. Interestingly, at T = 10 and T = 12, SCSO achieves marginally higher mean PSNR values than the hybrid approach. This can be attributed to SCSO’s aggressive exploration behavior, which occasionally finds sharper pixel-level intensity separations. However, this comes at the cost of higher variance and reduced structural similarity. In contrast, the hybrid SCSOWOA achieves more stable outputs, and its superiority in SSIM and FSIM confirms that it preserves structural and perceptual fidelity better, which is more clinically relevant for histopathological segmentation. The proposed hybrid algorithm also presents lower computational complexity than the compared techniques. This efficiency stems from its balanced design between exploration and exploitation and the incorporation of optimized search mechanisms. Reduced computational complexity increases the practicality of the algorithm for processing large-scale, high-resolution histopathological images. This reduction is particularly beneficial for real-time clinical decision support systems, where a balance between speed and accuracy is crucial. However, these findings do not indicate absolute superiority but rather reflect the algorithm’s relative performance under the specific dataset and parameter settings used. The performance of metaheuristic algorithms is known to be highly sensitive to parameter choices and initial conditions. For example, hyperparameters such as population size, number of iterations, and control parameters can significantly influence outcomes. Therefore, further experimental analyses are necessary to determine whether similar performance can be achieved under alternative configurations or datasets. Moreover, as image characteristics and threshold levels vary, the behavior of the algorithms may also change, which may pose a challenge to generalizability. Additionally, since each evaluation metric measures a different aspect of image quality, relying on a single metric for performance assessment may be methodologically insufficient. For instance, PSNR assesses only pixel-level similarity, while SSIM also incorporates structural information, potentially leading to contradictory results. This observation is corroborated by the relatively strong performance of the proposed hybrid method, where the higher Pearson and Spearman correlation values between the original and segmented pixels (0.9872 and 0.9923, respectively) suggest that the algorithm captures the overall image structure and pixel-level intensity distributions accurately. However, the moderate performance in Neighbor Pixel Pearson Correlation (0.8659) and Neighbor Pixel Spearman Correlation (0.8653) indicates that while the algorithm effectively preserves global structure, there may still be room for improvement in capturing finer, local details, particularly at the pixel neighborhood level. This could potentially lead to further improvements in segmentation, particularly in more complex image regions or where subtle variations in intensity exist. Despite this, the hybrid approach still outperforms others in terms of structural similarity as measured by SSIM and FSIM, which are more clinically relevant for histopathological applications, emphasizing the robustness and perceptual fidelity of the method. The relatively strong performance of the proposed hybrid method suggests that its adaptability and integration strategy have been effectively designed. However, this observation holds limited generalizability unless validated across different problem domains (e.g., medical image classification, denoising) and a wide variety of datasets. In addition to all these aspects, the clinical significance of the study lies not only in improving segmentation performance through numerical metrics but also in its potential use as a tool that can enhance diagnostic accuracy and reliability in the daily workflow of pathology experts. In this regard, SCSOWOA-based segmentation offers a viable approach that can be implemented as a decision support system in clinical practice.

## 6. Limitation

Although the proposed SCSOWOA-based hybrid optimization algorithm produces efficient results for the multilevel image thresholding problem in histopathological images, several limitations should not be overlooked. First, the performance evaluation is limited to a specific dataset involving lung cancer histopathological images. Variations in image structure, noise levels, and resolution may directly affect the optimization process and the output metrics. Second, the well-known parameter sensitivity issue associated with metaheuristic algorithms is only partially addressed in this study. The control parameters of the SCSO and WOA algorithms are kept constant, and no hyperparameter optimization is conducted. Consequently, the reported results are potentially contingent on these fixed parameter configurations. A more systematic parameter search or the adoption of adaptive control mechanisms could yield more robust performance in future studies. Another limitation lies in the reliance on only three objective image quality metrics—PSNR, SSIM, and FSIM—for performance evaluation. While these metrics capture different dimensions of image quality, subjective criteria such as human visual assessment are not included. These limitations should be carefully considered when interpreting and generalizing the findings, and future work should aim to address them through more comprehensive and diversified evaluations.

## 7. Conclusions and Future Works

Research on lung cancer segmentation using histopathological images remains limited in the literature, and existing approaches primarily rely on traditional image processing or deep learning-based techniques. This highlights the need for alternative methods capable of meaningfully processing high-resolution and structurally complex histopathological data. In this study, a novel adaptive optimization method is proposed by hybridizing the SCSO and WOA to solve the multilevel image thresholding problem in lung cancer histopathology. The proposed SCSOWOA optimizes threshold values over the image histogram, performing content-aware tissue separation rather than artificial partitioning. Experimental results show that the proposed method achieves notably high performance, particularly in structural similarity-based metrics such as SSIM and FSIM, indicating superior visual quality and structural preservation. The proposed SCSOWOA achieved 0.9340 SSIM and 0.9542 FSIM at T = 12, representing the most efficient structural and perceptual quality among all compared methods. The algorithm also reached 27.9453 dB average PSNR with low variance, and achieved an average execution time of 1.3221 s, which is up to 40% faster than conventional metaheuristic. These results confirm that the proposed hybrid approach provides a compelling balance of accuracy, stability, and efficiency for lung cancer histopathological image segmentation. This enables accurate segmentation of cancerous regions, offering a promising tool for early diagnosis and clinical decision support systems. The integration of the proposed model into the clinical workflow can be implemented as a decision support module integrated with digital pathology systems. For instance, following the digitization of histopathological slides during the scanning process, the algorithm can provide automatic segmentation, thereby reducing the workload of pathologists and highlighting suspicious regions. This approach may enhance clinical accuracy and time efficiency by supporting early diagnosis. In future studies, the integration of the algorithm with hospital information systems and image management platforms is planned. Nevertheless, due to parameter sensitivity and potential lack of generalizability across different datasets, further in-depth analyses are required. Future research directions include:Expanding test coverage by applying the method to diverse histopathological image sets and cancer types from various organs.Reducing parameter sensitivity through automated parameter tuning or the integration of adaptive control mechanisms.Enhancing computational efficiency via GPU-based acceleration and parallel processing techniques.Developing more advanced segmentation models by integrating with deep learning-based frameworks to construct hybrid systems.

## Figures and Tables

**Figure 1 diagnostics-16-00084-f001:**
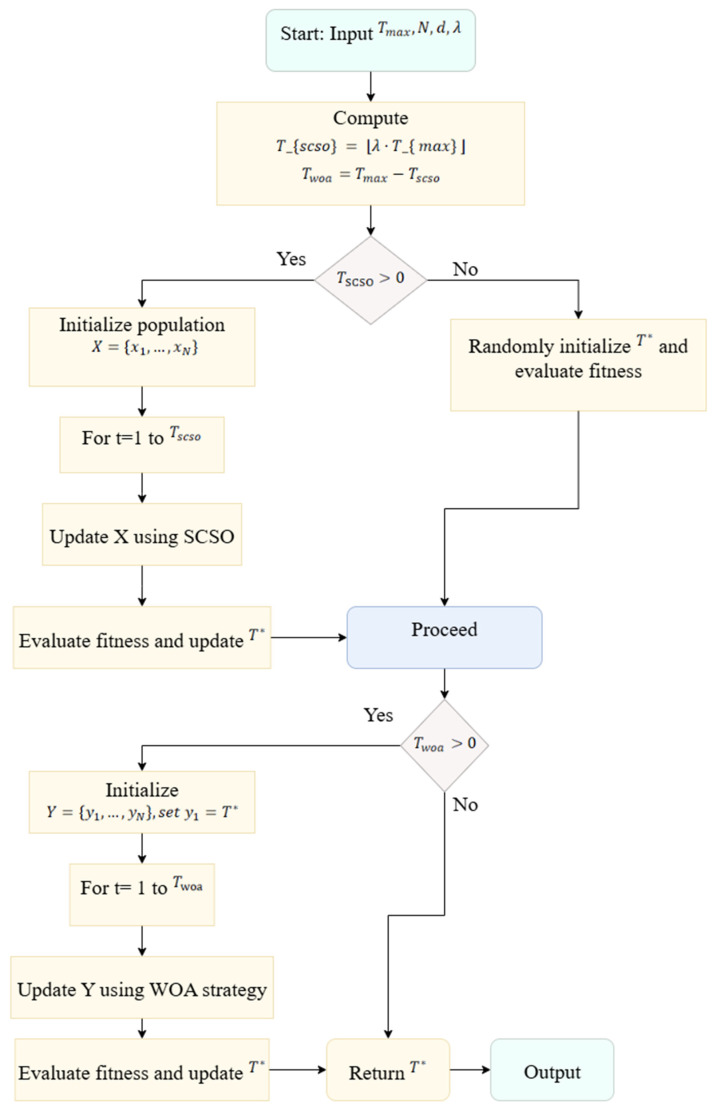
Flowchart of the proposed method.

**Figure 2 diagnostics-16-00084-f002:**
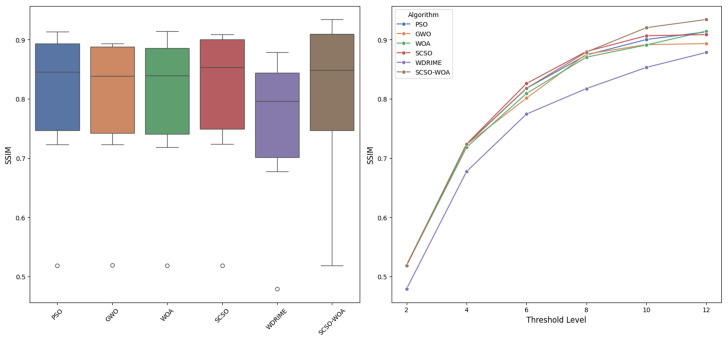
SSIM performance according to algorithms.

**Figure 3 diagnostics-16-00084-f003:**
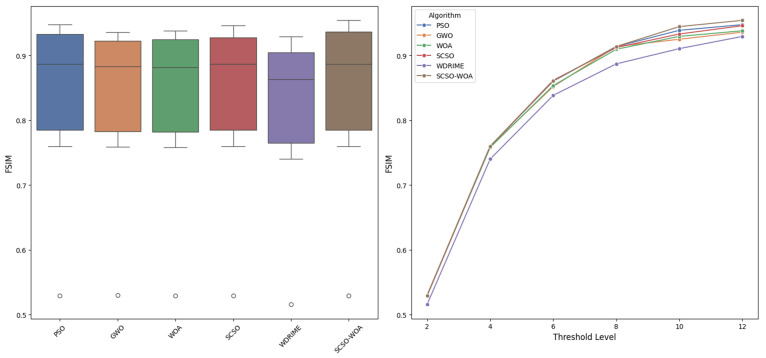
FSIM performance according to algorithms.

**Figure 4 diagnostics-16-00084-f004:**
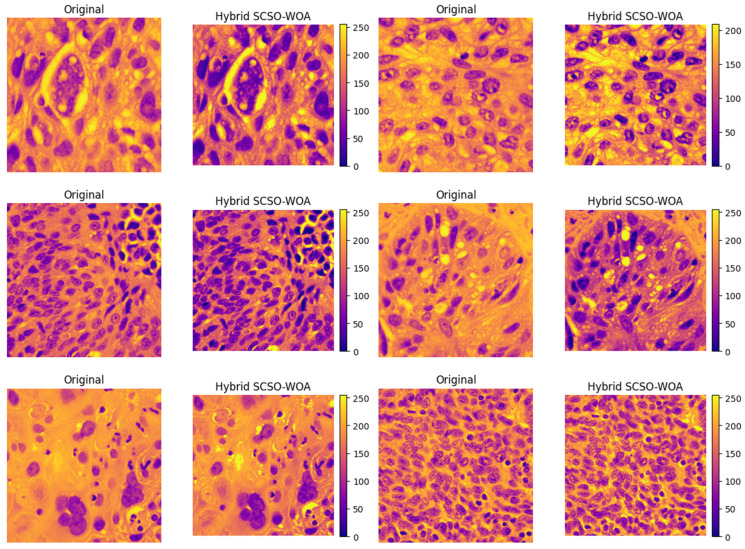
Visual outputs of test images.

**Figure 5 diagnostics-16-00084-f005:**
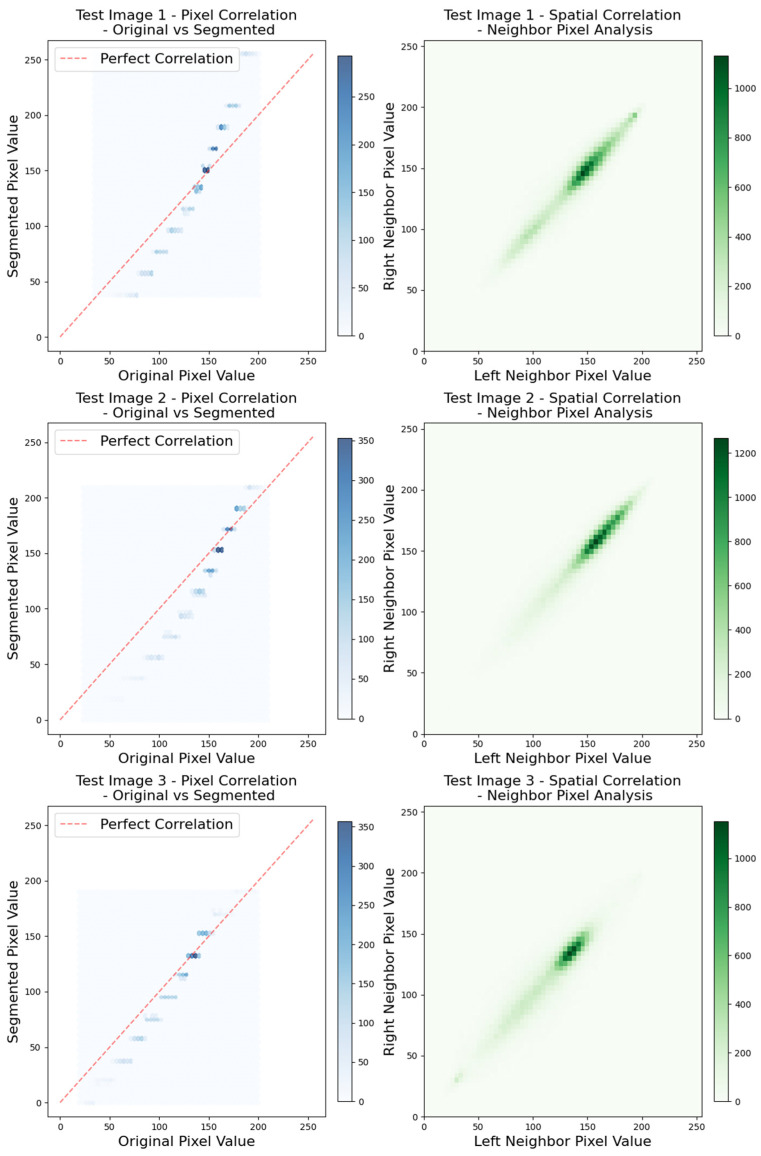
Spatial correlation and neighbor pixel analysis on test images.

**Table 1 diagnostics-16-00084-t001:** Parameter settings of the algorithms used.

Algorithm	Parameter
GWO	a: 2 → 0 (linear decreasing)
PSO	w: 0.5, c1: 2, c2: 2
WOA	a: 2 → 0 (linear decreasing)
SCSO	r: r_max_ → r_min_
WDRIME	WOA_Ratio_ = 0.4, Rime_Ratio_ = 0.4, DE_Ratio_ = 0.2, wf = 0.8, cr = 0.9
SCSOWOA (Proposed)	Switch_Ratio_ = 0.6

**Table 2 diagnostics-16-00084-t002:** Specifications of used software versions, hardware.

Specification	Details
Operating System	Windows 10—64 bit
Python Version	3.10
GPU	NVIDIA Tesla T4 GPU
CPU	i5-7300HQ CPU 2.50 GHz

**Table 3 diagnostics-16-00084-t003:** Summary statistics of PSNR according to algorithm and threshold level.

Algorithm	Threshold	PSNR	
		Mean	Standard Deviation
**PSO**	2	27.6058	0.0336
	4	27.5498	0.1203
	6	27.635	0.1630
	8	27.8011	0.0890
	10	27.7699	0.0932
	12	27.972	0.4338
**GWO**	2	27.6133	0.0428
	4	27.5642	0.1032
	6	27.5726	0.1083
	8	27.8229	0.0843
	10	28.2597	0.8699
	12	28.124	0.8018
**WOA**	2	27.6085	0.0336
	4	27.5414	0.0757
	6	27.6534	0.1659
	8	27.84	0.2200
	10	27.9709	0.3585
	12	28.4913	0.6677
**SCSO**	2	27.6058	0.0336
	4	27.5505	0.1183
	6	27.7327	0.1753
	8	28.1125	0.5049
	10	28.698	1.0292
	12	27.9561	0.6724
**WDRIME**	2	27.5932	0.0605
	4	27.5796	0.0837
	6	27.6063	0.0706
	8	27.5858	0.0568
	10	27.7394	0.4100
	12	27.7291	0.4013
**SCSOWOA (Proposed)**	2	27.6058	0.0336
	4	27.5549	0.1181
	6	27.6235	0.1002
	8	27.9977	0.2531
	10	28.6396	0.5454
	12	28.3346	0.6541

**Table 4 diagnostics-16-00084-t004:** Summary statistics of SSIM according to algorithm and threshold level.

Algorithm	Threshold	SSIM	
		Mean	Standard Deviation
**PSO**	2	0.5185	0.0739
	4	0.7226	0.0635
	6	0.8177	0.0463
	8	0.8740	0.0312
	10	0.9002	0.0259
	12	0.9131	0.0279
**GWO**	2	0.5194	0.0755
	4	0.7227	0.0631
	6	0.8012	0.0340
	8	0.8755	0.0365
	10	0.8916	0.0299
	12	0.8933	0.0551
**WOA**	2	0.5185	0.0739
	4	0.7179	0.0580
	6	0.8090	0.0328
	8	0.8704	0.0331
	10	0.8910	0.0323
	12	0.9144	0.0269
**SCSO**	2	0.5185	0.0739
	4	0.7236	0.0629
	6	0.8259	0.0395
	8	0.8800	0.0293
	10	0.9066	0.0389
	12	0.9085	0.0319
**Hybrid WDRIME**	2	0.4789	0.0783
	4	0.6771	0.0706
	6	0.7742	0.0587
	8	0.8173	0.0513
	10	0.8531	0.0502
	12	0.8783	0.0621
**SCSOWOA (Proposed)**	2	0.5185	0.0739
	4	0.7234	0.0635
	6	0.8177	0.0444
	8	0.8790	0.0340
	10	0.9202	0.0275
	12	0.9340	0.0156

**Table 5 diagnostics-16-00084-t005:** Summary statistics of FSIM according to algorithm and threshold level.

Algorithm	Threshold	FSIM	
		Mean	Standard Deviation
**PSO**	2	0.5295	0.0662
	4	0.7596	0.0470
	6	0.8603	0.0293
	8	0.9137	0.0181
	10	0.9388	0.0144
	12	0.9475	0.0091
**GWO**	2	0.5299	0.0668
	4	0.7593	0.0463
	6	0.8519	0.0221
	8	0.9138	0.0197
	10	0.9250	0.0210
	12	0.9360	0.0291
**WOA**	2	0.5295	0.0662
	4	0.7581	0.0450
	6	0.8535	0.0221
	8	0.9095	0.0199
	10	0.9294	0.0191
	12	0.9381	0.0130
**SCSO**	2	0.5295	0.0662
	4	0.7599	0.0466
	6	0.8616	0.0283
	8	0.9122	0.0170
	10	0.9331	0.0123
	12	0.9459	0.0102
**Hybrid WDRIME**	2	0.5159	0.0641
	4	0.7401	0.0512
	6	0.8384	0.0362
	8	0.8870	0.0264
	10	0.9105	0.0274
	12	0.9293	0.0302
**SCSOWOA (Proposed)**	2	0.5295	0.0662
	4	0.7598	0.0470
	6	0.8605	0.0282
	8	0.9138	0.0196
	10	0.9445	0.0113
	12	0.9542	0.0070

**Table 6 diagnostics-16-00084-t006:** Summary of overall algorithm performance.

Algorithm	PSNR		SSIM		FSIM	
	Mean	Std.	Mean	Std.	Mean	Std.
**PSO**	27.7591	0.1554	0.7989	0.0447	0.8349	0.0306
**GWO**	27.865	0.335	0.7950	0.0490	0.8310	0.0341
**WOA**	27.8273	0.2535	0.7946	0.0428	0.8295	0.0308
**SCSO**	28.0297	0.4222	0.8027	0.0460	0.8346	0.0301
**WDRIME**	27.6230	0.1804	0.7553	0.0618	0.8137	0.0392
**SCSOWOA**	27.9453	0.2840	0.8048	0.0431	0.8361	0.0298

**Table 7 diagnostics-16-00084-t007:** Comparison of average execution time of algorithms.

Algorithm	Execution Time Mean (s)
PSO	2.2134
GWO	2.0204
WOA	1.8831
SCSO	1.9132
Hybrid WDRIME [39]	1.9949
SCSOWOA (Proposed)	1.3221

**Table 8 diagnostics-16-00084-t008:** Performance comparison of the hybrid SCSO-WOA algorithm and U-Net on the LiTS17 dataset.

Algorithm	PSNR	SSIM	FSIM	Dice Coefficient	Jaccard Index	Hausdorff Measure
U-Net	9.02	0.1670	0.0004	0.3428	0.2089	152.49
SCSO-WOA	28.72	0.7011	0.7292	0.7211	0.5667	103.20

## Data Availability

Data is presented by the corresponding author based on request.

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
