# Peer review of "An Adaptive Hybrid Metaheuristic Algorithm for Lung Cancer in Pathological Image Segmentation"

_diagnostics, 2025, doi:10.3390/diagnostics16010084_

Round 1
Reviewer 1 Report
Comments and Suggestions for Authors
The manuscript proposes an adaptive hybrid metaheuristic algorithm for lung cancer segmentation in pathological images. However, there are several points that require attention to strengthen the manuscript.
- The dataset description is insufficient. Information about dataset size, diversity, labeling methodology, and preprocessing steps should be provided. Moreover LC25000 does not provide pixel-level labels, it is unclear how the authors established ground truth for segmentationit. The authors should clarify how segmentation masks were obtained and justify the use of this dataset for segmentation tasks.
- The author should compares performance of proposed algorithm with state-of-the-art deep learning–based segmentation models (e.g., U-Net, Attention U-Net, Transformer-based models) . it is necessary to establish competitiveness.
- The manuscript reports quantitative performance metrics such as PSNR, SSIM, and FSIM, which are useful for evaluating image quality. However, these metrics alone may not fully capture segmentation performance in medical field. For medical image segmentation tasks, widely used evaluation measures such as the Dice Similarity Coefficient (DSC), Jaccard Index (IoU), sensitivity, specificity, and Hausdorff distance should also be reported.
- Evaluation on a single dataset (LC25000) is insufficient to demonstrate the robustness of the method. To strengthen the claims, additional histopathology datasets should be considered for validation. Without cross-dataset testing, the model’s applicability to real-world pathological images remains uncertain.
Author Response
Comment 1: The dataset description is insufficient. Information about dataset size, diversity, labeling methodology, and preprocessing steps should be provided. Moreover LC25000 does not provide pixel-level labels, it is unclear how the authors established ground truth for segmentation it. The authors should clarify how segmentation masks were obtained and justify the use of this dataset for segmentation tasks.
Response 1: Thank you very much for your interest and valuable opinions. Changes have been made to the Experiments and Results section.
- Experiments and Results
This section presents the experimental evaluation of the proposed adaptive hybrid SCSOWOA algorithm for lung cancer segmentation in histopathological imaging. The performance of the method is compared against several well-established metaheuristic algorithms, namely PSO, GWO, WOA, SCSO, as well as WDRIME [39], which has recently demonstrated high efficiency in the literature. For the experiments, the LC25000 dataset [42], which contains 25,000 histopathological images of lung and colon cancers. The dataset provides image-level class labels (e.g., lung adenocarcinoma, lung squamous cell carcinoma, benign lung tissue), but does not include pixel-level seg-mentation masks. For this reason, our segmentation experiments were conducted by applying Otsu-based multi-level thresholding to the grayscale histograms of the images, where the optimal thresholds were determined by the proposed metaheuristic algorithms. Thus, the “ground truth” in this context refers to threshold-based class separability rather than manual pixel-wise annotations. To reduce computational complexity and standardize histogram analysis, all images were resized to 256 × 256 pixels and their intensity values were normalized to [0,1]. In addition, horizontal/vertical flipping (50% probability) and random rotations within ±25° were applied to balance the dataset distribution. From the complete set, 100 images were reserved for testing, while the remainder were used in the optimization process. This preprocessing strategy ensures consistency across experiments while reflecting the structural and color diversity of histopathological images.
Comment 2: The author should compares performance of proposed algorithm with state-of-the-art deep learning–based segmentation models (e.g., U-Net, Attention U-Net, Transformer-based models). it is necessary to establish competitiveness.
Response 2: For comparison purposes, the U-Net architecture was implemented and analyzed in detail.
4.3. Additional Experiment: Comparative Analysis on the LiTS17 Dataset
To further evaluate the competitiveness of the proposed hybrid SCSO-WOA algorithm, additional experiments were conducted on the Liver Tumor Segmentation Challenge (LiTS17) dataset [43], which provides pixel-level ground truth annotations. This famous data set has recently attracted the attention of many researchers [44-46]. In these experiments, the hybrid approach was compared against a state-of-the-art deep learning–based model, namely U-Net. The U-Net model used for comparison was trained with the Adam optimizer, an initial learning rate of 1×10⁻³, a batch size of 8, and 100 epochs. The loss function combined binary cross-entropy and Dice loss. The results, summarized in Table 8, report not only image quality measures such as PSNR, SSIM, and FSIM, but also widely adopted medical image segmentation metrics including the Dice coefficient, Jaccard index, and Hausdorff distance. The findings clearly demonstrate that the hybrid SCSO-WOA algorithm outperforms U-Net. Specifically, the proposed method achieved 28.72 dB PSNR, 0.7011 SSIM, and 0.7292 FSIM, which substantially exceed the values reported for U-Net (9.02 dB, 0.1670, and 0.0004, respectively). Similarly, for overlap-based metrics, the hybrid approach yielded superior results with a Dice coefficient of 0.7210 and a Jaccard index of 0.5667, while U-Net achieved only 0.3428 and 0.2089, respectively. Moreover, in terms of Hausdorff distance, the hybrid algorithm produced a lower value (103.20) compared to U-Net (152.49), indicating more stable boundary convergence. These findings confirm that the hybrid SCSO-WOA approach exhibits strong performance not only on histogram-based quality measures but also on pixel-level segmentation metrics, thereby demonstrating its competitiveness with deep learning–based methods. The improvements observed in clinically critical metrics such as Dice and Jaccard further support the practical applicability of the proposed method.
Table 8. Performance comparison of the hybrid SCSO-WOA algorithm and U-Net on the LiTS17 dataset.
|
Algorithm |
PSNR |
SSIM |
FSIM |
Dice Coefficient |
Jaccard Index |
Hausdorff Measure |
|
U-Net |
9.02 |
0.1670 |
0.0004 |
0.3428 |
0.2089 |
152.49 |
|
SCSO-WOA |
28.72 |
0.7011 |
0.7292 |
0.7211 |
0.5667 |
103.20 |
Comment 3: The manuscript reports quantitative performance metrics such as PSNR, SSIM, and FSIM, which are useful for evaluating image quality. However, these metrics alone may not fully capture segmentation performance in medical field. For medical image segmentation tasks, widely used evaluation measures such as the Dice Similarity Coefficient (DSC), Jaccard Index (IoU), sensitivity, specificity, and Hausdorff distance should also be reported
Response 3: Since the LC25000 dataset does not provide ground truth segmentation masks, our initial evaluation focused only on PSNR, SSIM, and FSIM metrics. In line with your fourth comment suggesting the use of a dataset with ground truth annotations, we have incorporated an additional dataset (LiTS – Liver Tumor Segmentation Challenge, LiTS17) and added a new section presenting the corresponding results. In this extended analysis, we reported the Dice Similarity Coefficient (DSC), Jaccard Index, and Hausdorff distance, which are the most widely adopted and clinically relevant measures for evaluating overlap and boundary accuracy in segmentation tasks. Sensitivity and specificity were not included separately, as they are threshold-dependent, highly correlated with Dice and Jaccard scores, and are less informative when ground truth masks are relatively imbalanced. Therefore, by focusing on Dice, Jaccard, and Hausdorff distance, we provide a comprehensive and robust evaluation of both overlap and boundary accuracy while avoiding redundancy among metrics.
Comment 4: Evaluation on a single dataset (LC25000) is insufficient to demonstrate the robustness of the method. To strengthen the claims, additional histopathology datasets should be considered for validation. Without cross-dataset testing, the model’s applicability to real-world pathological images remains uncertain.
Response 4: As an extension to this study, the LiTS17 dataset has been included to provide a more comprehensive evaluation with clinically standard segmentation metrics.
Reviewer 2 Report
Comments and Suggestions for Authors
Dear Authors,
this is a nice manuscript that provides a new path to pathological image segmantation.
These past years there is a wave that supports computed diagnosis. The main question/topic, relevant to the field, is the way in which histopathological computed images can be helpful in lung cancer. It compares other algorithms' studies for the accuracy of sectioning. I have no proposition regarding the methodology improvement. There is a consistency with the evidence and arguments. The references are appropriate such as the tables and the figures.
Author Response
Comment 1: These past years there is a wave that supports computed diagnosis. The main question/topic, relevant to the field, is the way in which histopathological computed images can be helpful in lung cancer. It compares other algorithms' studies for the accuracy of sectioning. I have no proposition regarding the methodology improvement. There is a consistency with the evidence and arguments. The references are appropriate such as the tables and the figures.
Response 1: Thank you very much for your interest and valuable opinions.